# The Efficacy and Safety of Vonoprazan and Amoxicillin Dual Therapy for *Helicobacter pylori* Infection: A Systematic Review and Network Meta-Analysis

**DOI:** 10.3390/antibiotics12020346

**Published:** 2023-02-07

**Authors:** Ligang Liu, Fang Li, Hekai Shi, Milap C. Nahata

**Affiliations:** 1Institute of Therapeutic Innovations and Outcomes (ITIO), College of Pharmacy, The Ohio State University, Columbus, OH 43210, USA; 2Department of Pharmacy, Beijing You An Hospital, Capital Medical University, Beijing 100069, China; 3Department of General Surgery, Fudan University Affiliated Huadong Hospital, Shanghai 200040, China; 4Colleges of Pharmacy and Medicine, The Ohio State University, Columbus, OH 43210, USA

**Keywords:** *Helicobacter pylori*, vonoprazan dual therapy, antibiotic resistance, eradication therapy, network meta-analysis

## Abstract

The eradication of *Helicobacter pylori (H. pylori)* infection remains challenging due to increasing bacterial resistance. Resistance rates to clarithromycin, metronidazole, and levofloxacin were higher than 30% in the USA, making current therapies less effective. Vonoprazan triple therapy (VAC) has demonstrated similar efficacy and safety profiles compared to PPI-based triple therapy (PPI). However, the eradication rate of vonoprazan dual therapy (VA) for *H. pylori* infection in comparison to VAC, and PPI was poorly established. Electronic databases were searched up to 6 October 2022, to identify studies examining the safety and efficacy of VA compared to VAC and PPI. Six studies were included. For empiric therapies among treatment naïve patients, VA, VAC, and PPI did not achieve high cure rates (>90%). The comparative efficacy ranking showed VAC was the most effective therapy, followed by VA, and PPI. The results were similar for clarithromycin-resistant infections. The comparative safety ranking showed VA ranked first, whereas PPI triple therapy was the least safe regimen. These findings should guide the selection of the most effective and safe treatment and conduct additional studies to determine the place of vonoprazan dual versus triple therapies in patients with *H. pylori* from various countries across the world.

## 1. Introduction

*Helicobacter pylori (H. pylori)* infection is one of the most common chronic infections across the world. Approximately 50% of the population may test positive for *H. pylori* infection worldwide [1]. *H. pylori* infection is associated with chronic gastritis, peptic ulcer disease, gastric mucosa-associated lymphoid tissue lymphoma, and gastric cancer [2], which may cost billions of dollars annually in the United States [3].

*H. pylori* has been classified as a top carcinogen by the World Health Organization (WHO) since 1994 and was listed as a human carcinogen by the U.S. Department of Health and Human Services in 2021 [4,5]. *H. pylori* infection is the most important modifiable risk factor for preventing gastric cancer, and successful eradication can reduce the risk of gastric cancer [6]. It should be noted that *H. pylori* infection generally does not disappear spontaneously and may lead to lifelong infection [7]. Thus, medication therapy should be used to eradicate *H. pylori* to reduce the risk of gastric cancer and decrease the mortality rate [8].

In the US, clarithromycin triple therapy with a proton pump inhibitor (PPI), bismuth quadruple therapy, concomitant therapy, sequential therapy, hybrid therapy, and levofloxacin triple or sequential therapy have been suggested as first-line therapies for the treatment of *H. pylori* infection [9]. However, the eradication of *H. pylori* infection remains challenging, notably due to increasing bacterial resistance [10]. Resistance rates to clarithromycin, metronidazole, and levofloxacin were higher than 30% in the USA [11]. Rokkas et al. reported that most PPI-based therapies failed to achieve the desired 90% eradication rate [12]. Some characteristics of PPIs, including short elimination half-lives, insufficient acid suppression, and pharmacokinetic differences among different races, may make PPIs less than an ideal regimen worldwide [13]. Treatment-emergent adverse events can negatively impact adherence, limiting overall efficacy as well [14]. Therefore, an important priority has been to explore new and effective eradication treatments for *H. pylori* infections.

As a novel potassium-competitive acid blocker (P-CAB), vonoprazan (VPZ) produced a complete acid-inhibition effect with a rapid onset, sustaining pH at the targeted level (6–7) for over 24 h, and undergoing metabolism without CYP2C19 polymorphisms [15]. Vonoprazan has been used for the treatment of *H. pylori* infection in Japan since 2014 [16]. Vonoprazan-based therapy was the standard therapy in Japanese guidelines [17]. In the Maastricht VI/Florence consensus report, vonoprazan in combination with antibiotics was also recommended as a first-line and second-line treatment, especially in patients with evidence of antimicrobial-resistant infections [18]. Vonoprazan triple therapy has demonstrated comparable or favorable efficacy and safety profiles compared to PPI-based triple therapy [19,20,21,22]. However, the efficacy and safety of vonoprazan dual therapy relative to vonoprazan triple therapy and PPI triple therapy are still unknown. In this network meta-analysis (NWM), we aimed to assess the comparative eradication rate and safety of vonoprazan dual therapy, vonoprazan triple therapy, and PPI triple therapy in adults with *H. pylori* infection.

## 2. Method

### 2.1. Search Strategy and Selection Criteria

This network meta-analysis was performed using the Preferred Reporting Items for Systematic review and Meta-Analysis 2020 (PRISMA) statement [23]. The study protocol was registered on PROSPERO with registration number CRD42022357726. Electronic databases, including PubMed, Embase, Cochrane Library, and Scopus, were searched without any date limitations up to 6 october 2022. The following terms were used to conduct the searches: dual, “*Helicobacter pylori*”, “*H. pylori*”, vonoprazan, “potassium-competitive acid blocker”, P-CAB, and TAK-438.

Studies were included in this NWM if they met all criteria: (1) adult diagnosed with *H. pylori* infection, (2) studies using VPZ-based dual therapy as the interventional group and PPI-based or VPZ-based triple therapies as the control group, and (3) eradication rate included in the outcomes. Studies were excluded if they met any of the following criteria: (1) animal studies, (2) single-arm studies, (3) studies published in languages other than English and Chinese, (4) meta-analysis or systematic review of other studies, (5) review articles, (6) case reports or case series, (7) conference abstracts or journal abstracts.

### 2.2. Outcome Measures

The primary outcome was the eradication rate based on an intention-to-treat (ITT) analysis and/or per protocol (PP) analysis. Secondary outcomes included dropout rates and commonly reported adverse events.

### 2.3. Data Extraction and Quality Assessment

Two authors (L.L. and L.F.) independently reviewed all articles by reading the titles and abstracts for eligibility. After screening all articles, two reviewers (L.L. and L.F.) independently extracted the data, including the name of the first author, publication year, country, study design, treatment regimens, duration of treatment, the method used to confirm successful eradication, eradication rate (ITT and/or PP analysis), dropout rate, and adverse events. A third reviewer (M.N.) would be consulted if any disagreement existed.

Treatment regimens were classified into three groups: (1) vonoprazan dual therapy (VA, vonoprazan and amoxicillin), (2) vonoprazan triple therapy (VAC, vonoprazan, amoxicillin, and clarithromycin), and (3) PPI triple therapy (a PPI, amoxicillin, and clarithromycin). Two authors (L.L. and L.F.) independently assessed the risk of bias of the included studies. The Cochrane tool for assessing Risk of Bias in randomized trials version 2 (RoB V.2.0 tool) was used to assess the quality of randomized trials [24]. The quality of cohort studies was assessed using the Newcastle–Ottawa Scale quality instrument. Any disagreements among the first three authors were discussed with the last author (M.N.).

### 2.4. Data Synthesis and Statistical Analysis

R software (GitHub, San Francisco, CA, USA; version 4.1.2) with the “gemtc” package (version 1.0–1) was used for statistical analysis. The actual cure rates of VPZ-based dual therapy, VPZ-based triple therapy, and PPI-based triple therapy were presented as pooled eradication rates with a 95% confidence interval (95% CI). We compared the following data between the three groups using odds ratios (ORs) with a 95% credible interval (95% CrI): (1) the eradication rate, (2) the dropout rate, (3) the incidence of total adverse events, and (4) each adverse event reported. The I2 test was used to estimate the heterogeneity, and an I^2^ < 25% indicated low heterogeneity. In all analyses, *p* < 0.05 (two-tailed) or 95% CrI of OR excluding 1 was considered statistically significant in all analyses. Surface under the cumulative ranking curve (SUCRA) values were used to examine the probability of cumulative ranking. The higher the SUCRA value, the more effective and safer the regimen was. Comparison-adjusted funnel plots were used to examine potential publication biases.

### 2.5. Subgroup Analysis

We performed several subgroup analyses based upon the duration of treatment (7 days or 14 days), types of studies (clinical trials or observational studies), different amoxicillin doses (1500 mg daily or >1500 mg daily), and the geographic location (Asia or non-Asia).

## 3. Results

### 3.1. Study Characteristics and Quality Assessment

A total of 167 studies were identified after a comprehensive literature search. 89 duplicate records were removed. After the title and abstract screening, 56 studies were excluded. 22 studies were reviewed as full text and six studies were included in the final analysis [25,26,27,28,29,30] (Figure 1).

Among 6 studies included, three were conducted in Japan, one in Pakistan, one in the United States and Europe, and one in China. Five studies were randomized clinical trials, with one being a retrospective cohort study. The majority of RCTs were open-label trials. In all studies, 20 mg of vonoprazan was given twice daily to all participants. The amoxicillin dose ranged from 1500 to 3000 mg daily. The dose of clarithromycin was 200 mg twice daily in the studies conducted in Japan, which was in line with the Japanese guidelines for the treatment of *H. pylori*, and the dose was 500 mg twice daily in studies conducted in other countries. Lansoprazole and omeprazole were the PPIs used in these trials. The duration of therapy was 7 days in most included studies. The urea breath test was used to ensure the successful eradication of *H. pylori* infection in most studies. Four studies provided the eradication rate in both ITT and PP analyses, and two studies only reported the eradication rate in PP analysis. Therefore, we used the eradication rate from the PP analysis in this NWM. The characteristics of the involved studies are presented in Table 1.

Overall, 1694 patients contributed to the efficacy analysis and 1922 to the safety analysis. In the PP analysis, 750 patients received vonoprazan dual therapy, and the eradication rate was 85.9% (95% CI, 76.5% to 92.0%); 580 patients were in the vonoprazan triple therapy group, and the eradication rate was 87.4% (95%CI, 77.3% to 93.4%); 364 patients had PPI triple therapy, and the eradication rate was 76.5% (95%CI, 65.0% to 85.1%).

The risk of bias evaluation is shown in Table 2. The studies included in this network meta-analysis were of good quality. Overall, all randomized controlled trials had a low risk for bias. The observational study was of good quality, with seven stars.

### 3.2. Primary Outcome: Efficacy

The network map of the three therapeutic interventions is described in Figure 2. The node size reflected the number of patients who received each treatment. The most common comparison was vonoprazan dual therapy and vonoprazan triple therapy. Compared with PPI triple therapy, vonoprazan dual therapy (OR = 2.04, 95% Crl 1.15 to 4.07) and triple treatment (OR = 2.54, 95% Crl 1.39 to 5.46) showed a significantly higher eradication rate. Vonoprazan dual therapy did not provide greater efficacy than vonoprazan triple therapy (OR = 0.80, 95% Crl 0.49 to 1.29) (Table 3). Judging by SUCRA values, vonoprazan triple therapy (SUCRA = 92.6%) was the best treatment, followed by vonoprazan dual therapy (SUCRA = 56.3%), and PPI-based therapy (SUCRA = 1.1%). The funnel plot of the studies was found to be symmetrical (Figure 3), indicating that there was no publication bias or small sample size study effects.

### 3.3. Subgroup Analysis

#### 3.3.1. Study Type Effect

To explore the effect of types of studies on the eradication rate, 5 randomized clinical trials were placed as one group. VA therapy was not associated with a higher efficacy (OR = 0.77, 95% Crl 0.43 to 1.30) compared to VAC therapy and had a significantly higher eradication rate than PPI-based triple therapy (OR = 2.04, 95% Crl 1.04 to 4.42) (Table 3). 

#### 3.3.2. Treatment Duration Effect

We divided the studies into two groups based on treatment duration. In the 7-day duration group, vonoprazan dual therapy did not show a significantly higher effect than both PPI triple therapy and vonoprazan triple therapy (OR = 2.07, 95% Crl 0.86 to 5.66). In the 14-day group, vonoprazan dual therapy showed a comparable effect to vonoprazan triple therapy (OR = 0.84, 95% Crl 0.38 to 1.75) (Table 3).

#### 3.3.3. Amoxicillin Dose Effect

To explore the effect of amoxicillin daily dose on eradication rate, studies with a total daily dose of 1500 mg versus greater than 1500 mg were placed into two groups. In the high-dose group, vonoprazan dual therapy showed significantly higher efficacy than PPI-based therapy (OR = 2.13, 95% Crl 1.03 to 4.86), and comparative efficacy to vonoprazan triple therapy (OR = 0.85, 95% Crl 0.44 to 1.78). Vonoprazan dual therapy did not offer higher efficacy than vonoprazan triple therapy in the low-dose group as well (OR = 0.71, 95% Crl 0.23 to 2.07) (Table 3).

#### 3.3.4. Regional Effect

Since 5 of 6 studies were conducted in various Asian countries, we grouped the studies from Asia into one group. Vonoprazan dual therapy was not associated with a greater effect than vonoprazan triple therapy (OR = 0.83, 95% Crl 0.38 to 1.68) or PPI triple therapy (OR = 2.86, 95% Crl 0.63 to 13.29) (Table 3).

#### 3.3.5. Efficacy of Clarithromycin-Resistant Strains

Compared with VPZ triple therapy and PPI triple therapy, VPZ dual therapy did not demonstrate higher efficacy (OR = 0.84, 95% Crl 0.55 to 1.31; OR = 1.57, 95% Crl 0.82 to 3.01, separately). Similarly, the rank probability results showed that VPZ triple therapy had the best performance (SUCRA = 89.3%), followed by VPZ dual therapy (SUCRA = 56.3%), and PPI triple therapy (SUCRA = 4.4%).

### 3.4. Secondary Outcomes

#### 3.4.1. Treatment Discontinuation

Compared with PPI-based triple therapy, a significant difference in discontinuation of therapy was not observed in the VAC group (OR = 1.24, 95% Crl 0.11 to 12.07) and VA group (OR = 1.49, 95% Crl 0.17 to 19.09). VA group did not show a significant difference versus the VAC group as well (OR = 1.21, 95% Crl 0.32 to 6.78) (Table 4). VA had a higher likelihood of discontinuation of therapy (SUCRA = 36.9%) compared to VAC (SUCRA = 51.3%), and PPI triple therapy (SUCRA = 61.8%) (Table 5).

#### 3.4.2. Total Incidence of Adverse Events

Vonoprazan dual therapy was not associated with a lower risk of adverse events compared to vonoprazan triple therapy (OR = 0.69, 95% Crl 0.36 to 1.24), and PPI triple therapy (OR = 0.52, 95% Crl 0.20 to 1.08) (Table 4). Nevertheless, vonoprazan dual therapy had a higher probability of fewer adverse events (SUCRA = 93.8%), followed by vonoprazan triple therapy (SUCRA = 43.3%), and PPI triple therapy (SUCRA = 12.8%) (Table 5).

#### 3.4.3. Diarrhea

Vonoprazan dual therapy had a comparable effect of being associated with diarrhea relative to vonoprazan triple therapy (OR = 0.70, 95% Crl 0.33 to 1.34), and a lower incidence of diarrhea than PPI triple therapy. (OR = 0.36, 95% Crl 0.14 to 0.85) (Table 4). The rank probability results favored VA (SUCRA = 93.1%), followed by VAC (SUCRA = 53.1%), and PPI (SUCRA = 37.4%) (Table 5).

#### 3.4.4. Abdominal Pain

Vonoprazan dual therapy did not have statistically significant differences associated with abdominal pain compared to vonoprazan triple therapy (OR = 1.09, 95% Crl 0.34, 3.52), and PPI-based triple therapy (OR = 0.66, 95% Crl 0.10 to 4.07) (Table 4). Vonoprazan triple therapy showed the best performance (SUCRA = 64.2%), followed by vonoprazan dual therapy (SUCRA = 57.0%), and PPI triple therapy (SUCRA = 28.8%) (Table 5).

#### 3.4.5. Bloating

Vonoprazan dual therapy did not show a lower incidence of bloating compared to vonoprazan triple therapy (OR = 1.48, 95% Crl 0.34 to 6.67), and PPI triple therapy (OR = 0.30, 95% Crl 0.04 to 1.70) (Table 4). The rank probability results favored vonoprazan triple therapy (SUCRA = 83.4%), followed by vonoprazan dual therapy (SUCRA = 59.2%), and PPI triple therapy (SUCRA = 7.50%) (Table 5).

#### 3.4.6. Constipation

Relative to VAC, VA had a comparable effect on the incidence of constipation (OR = 1.76, 95% Crl 0.51 to 7.46) (Table 4). VAC showed a lower risk of constipation (SUCRA = 82.2%) compared to VA (SUCRA = 17.8%) (Table 5).

## 4. Discussion

Even though *H. pylori* infection affects billions of individuals worldwide, the best treatment remains uncertain. The increasing resistance to clarithromycin and metronidazole has been alarming, leading to treatment failures [31,32]. Traditional PPI-based triple therapy has been used for decades, and the eradication rate has been reported to be as low as 69.6% [33]. Compared with PPI, vonoprazan has a more potent acid inhibition effect, and a favorable pharmacokinetic profile [34,35]. A previous meta-analysis showed that vonoprazan triple therapy achieved an eradication rate of greater than 90% and exceeded the eradication rate observed with PPI-based therapy [12,19]. However, the efficacy and safety profile of vonoprazan dual therapy compared with other treatments have not been well established.

This network meta-analysis evaluated the efficacy and safety of vonoprazan dual therapy compared with vonoprazan triple therapy and PPI triple therapy. The rank probability results favored vonoprazan triple therapy over other interventions for the eradication rate. Similar trends were observed for clarithromycin-resistant infections. As for safety, the rank probability results demonstrated that VA therapy seemed to decrease the probability of the incidence of total adverse events, diarrhea, and dysgeusia. The administration of broad-spectrum antimicrobials can profoundly damage the structure and decrease the diversity of intestinal microbiota, causing severe diarrhea [36]. Vonoprazan triple therapy was associated with a lower probability of abdominal pain, bloating, constipation, and nausea/vomiting. PPI triple therapy showed a higher likelihood of all adverse effects. It is important to note that vonoprazan dual therapy had a significantly decreased risk of diarrhea compared to the other two regimens, each containing two antibiotics, thus increasing the risk of antibiotic-associated diarrhea [37].

The subgroup analysis revealed that the duration of therapy, types of studies, and regional locations of the included studies did not affect the eradication rate. However, with a higher amoxicillin dose, vonoprazan dual therapy had a greater efficacy outcome than PPI-based therapy. Due to the lack of comparison with PPI triple therapy in the low-dose amoxicillin group, there was not enough evidence to conclude that high-dose amoxicillin may contribute to higher efficacy. However, PPI dual therapy with high-dose amoxicillin has demonstrated significant improvement in efficacy, even higher than 90% [38]. These data emphasize the need to conduct studies to determine the eradication rate with an increased amoxicillin dose in vonoprazan dual therapy.

Graham et al. suggested that the duration of therapy influences the eradication of *H. pylori* infection [39]. An extended duration may be able to overcome persistent bacteria. A recent systematic review concluded that a prolonged duration of traditional PPI-based triple therapy to 14 days was associated with higher eradication rates [40]. Therefore, 14 days of therapy is the preferred duration over 7 days for certain patients with *H. pylori* infection.

It is important to note that all participants included in this network meta-analysis were treatment naïve patients. Hu et al. investigated the eradication rate of vonoprazan dual therapy in treatment naïve patients using different doses of amoxicillin and for different durations, and none of the combinations provided an eradication rate greater than 90% [41]. Gotoda et al. also found that in treatment naïve patients, the eradication rate of vonoprazan dual therapy did not reach 90% [42]. Further, a recent meta-analysis showed that the eradication rate of vonoprazan dual therapy was less than 90% [43]. Based on these data, vonoprazan dual therapy may not be preferred as an initial treatment because of the relatively low eradication rate. Gao et al. reported the eradication rate of vonoprazan with 3000 mg amoxicillin daily for 14 days, achieving an eradication rate of 92.5% in patients who had failed previous regimens [44]. This may indicate that the use of vonoprazan and amoxicillin high-dose therapy is preferred in patients with previous treatment failure.

To address the challenges of refractory infections, several strategies may be used to improve the efficacy and safety of treatments. Patient education had positive effects on the *H. pylori* eradication rate and medication adherence [45]. Probiotics, in conjunction with recommended therapies, slightly increased the eradication rate and significantly decreased the adverse effects of antibiotics [46]. Susceptibility testing may guide the selection of the most effective treatment regimens and improve the eradication rate to exceed 90% [47]. It has been suggested that amoxicillin, clarithromycin, levofloxacin, and metronidazole should be routinely evaluated for *H. pylori* susceptibility [48,49]. Tailored therapy also had better outcomes than empirical therapy for *H. pylori* infection [50]. Moreover, vonoprazan-amoxicillin dual therapy with adjustment of amoxicillin dose based on body size may minimize antibiotic resistance development and gut microbiota dysbiosis, while providing the desired efficacy for eradication [51].

It has been suggested that vonoprazan-based triple therapy may contribute to increased global antimicrobial resistance [52]. The use of clarithromycin can drive antibiotic resistance at the population level [53] and would not be effective for the treatment of clarithromycin-resistant infections [54]. Therefore, Suzuki et al. suggested the use of vonoprazan dual therapy because it may provide comparable eradication rates, improve safety and tolerability, and minimize antimicrobial resistance [55]. Moreover, ACG guidelines recommend avoiding clarithromycin in locations where the resistance rate to clarithromycin is greater than 15% [9]. Therefore, before starting vonoprazan triple therapy, the local resistance rate to clarithromycin should be referenced, especially in the US, where the resistance rate exceeds 30%. Ouyang et al. showed that vonoprazan dual therapy had comparable efficacy and safety profiles compared with vonoprazan triple therapy [56]. However, several journal abstracts instead of original articles were included in their meta-analysis, which could have led to potential bias. Further, the optimization of the duration of therapy and the optimal amoxicillin dosing with vonoprazan dual therapy should be studied in future research.

This network meta-analysis has several limitations. Only six studies could be included in this analysis since vonoprazan therapies have not been widely available across the world for substantial time. Most studies were conducted in Asia, with one in the US. Additional studies should be done in Western countries due to geographic differences in antibiotic resistance patterns for *H. pylori* infection and potential variability in population characteristics. In addition, a comparison between other available treatments, such as bismuth quadruple therapy with vonoprazan dual therapy, has not been performed at this time. Future research should compare regimens with vonoprazan dual therapy and vonoprazan triple therapy to identify the potential use of vonoprazan dual versus triple therapy for the treatment of *H. pylori* infection.

## 5. Conclusions

For empiric therapies of *H. pylori* infection among treatment naïve patients, vonoprazan dual therapy, vonoprazan triple therapy, and PPI triple therapy did not achieve high cure rates (>90%). The comparative efficacy ranking based on SUCRA values showed that vonoprazan triple therapy was the most effective, and PPI triple therapy the least effective regimen. The comparative safety ranking showed that vonoprazan dual therapy was the safest regimen compared to the other two regimens. However, additional clinical trials and real-world studies need to be conducted in Western countries to determine the specific role of vonoprazan dual versus triple therapy based on their relative efficacy and safety in patients with *H. pylori* infection in various parts of the world.

## Figures and Tables

**Figure 1 antibiotics-12-00346-f001:**
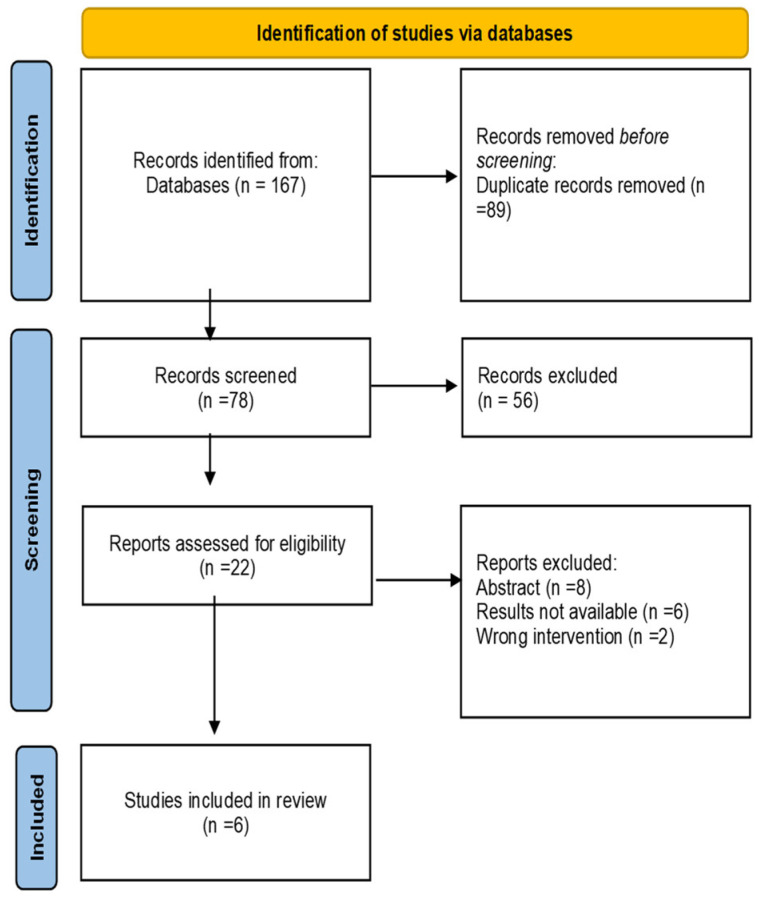
Preferred Reporting Items for Systematic Reviews and Meta-Analyses Flow Diagram.

**Figure 2 antibiotics-12-00346-f002:**
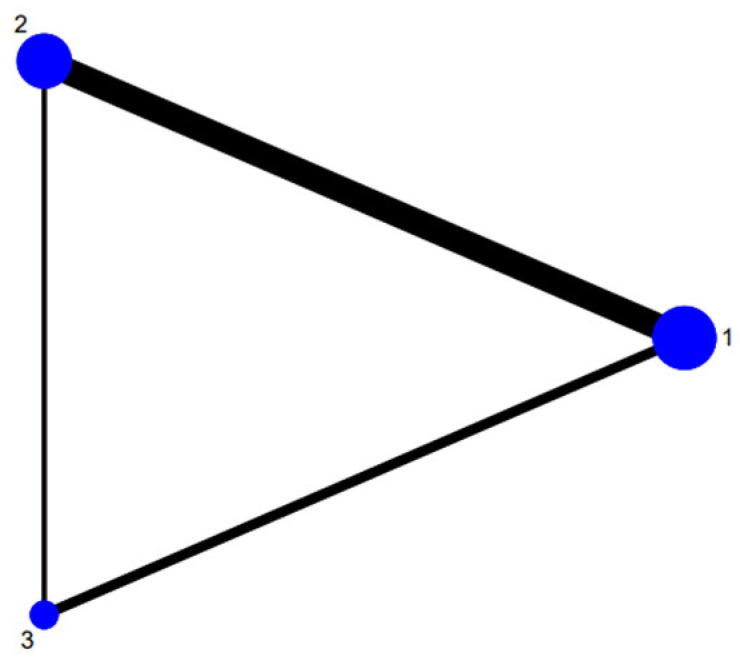
Network of included studies with available direct comparisons. 1. Vonoprazan-amoxicillin; 2. vonoprazan-amoxicillin-clarithromycin; 3. PPI-amoxicillin-clarithromycin.

**Figure 3 antibiotics-12-00346-f003:**
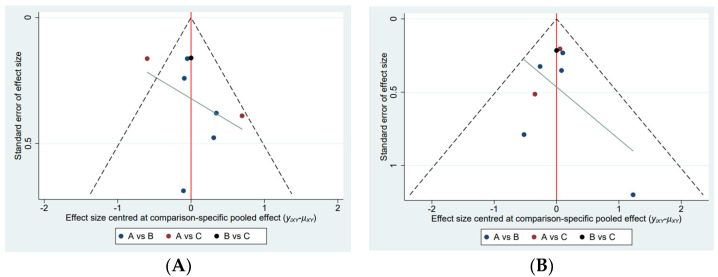
Comparison−adjusted funnel plot. (**A**) Adverse events, (**B**) Efficacy A, Vonoprazan dual therapy; B, vonoprazan triple therapy; C, PPI triple therapy.

**Table 1 antibiotics-12-00346-t001:** Characteristics of studies included in the network meta-analysis.

Authors (Year, Country)	Study Design	Number	Test for Confirming Helicobacter Pylori		Duration	Eradication Regimen		Eradication Rate		Common Adverse Events
			Infection	Eradication		VA	VAC/PPI	ITT	PP	
Furuta et al. (2020, Japan) [25]	Retrospective cohort study	112	RUT	UBT	7 days	VA: VPZ 20 mg bid, AMO 500 mg tid	VAC: VPZ 20 mg, AMO 750 mg, CLA 200 mg, bid	VA: 92.9% (52/56), VAC: 91.9% (51/56), *p* = 0.728	VA: 94.4% (51/54), VAC: 92.7% (51/55), *p* = 0.715	Diarrhea: VA (8.9%); VAC (17.8%)
										Nausea/vomiting: VA (0%); VAC (1.8%)
										Skin rash: VA (3.6%); VAC (0%)
										Abdominal pain: VA (1.8%); VAC (3.6%)
										Dysgeusia: VA (0%); VAC (1.8%)
Suzuki et al. (2020, Japan) [28]	RCT	335	C	UBT	7 days	VA: VPZ 20 mg, AMO 750 mg, bid	VAC: VPZ 20 mg, AMO 750 mg, CLA 200 mg, bid	VA: 84.5% (142/168); VAC: 89.2% (149/167), *p* = 0.203	VA: 87.1% (142/163); VAC: 90.2% (148/164), *p* = 0.372; CLA-R VA: 92.3%, VAC: 76.2%, *p* = 0.048	Total: VA (27.4%); VAC (30.5%), *p* = 0.524
										Diarrhea: VA (9.5%); VAC (15.0%)
										Nausea/vomiting: VA (3.0%); VAC (2.4%)
										Skin rash: VA (5.4%); VAC (3.0%)
										Abdominal pain: VA (3.6%); VAC (1.2%)
										Dysgeusia: VA (1.8%); VAC (1.8%)
										Bloating: VA (11.9%); VAC (8.4%)
Horii et al. (2021, Japan) [27]	RCT	43	C	C	7 days	VA: VPZ 20 mg, AMO 750 mg, bid	VAC: VPZ 20 mg, AMO 750 mg, CLA 200 mg, bid	N/A	VA: 84.2% (16/19); VAC: 95.8% (23/24), *p* = 0.31	Diarrhea: VA (5.3%); VAC (12.5%)
										Abdominal pain: VA (5.3%); VAC (0)
										Dysgeusia: VA (0); VAC (4.2%)
										Skin rash: VA (5.3%); VAC (0)
										Nausea: VA (10.5%); VAC (0)
Zuberi et al. (2022, Pakistan) [29]	RCT	179	SAT or Giemsa Stain	HP Stool Ag	14 days	VA: VPZ 20 mg, AMO 1g, bid	OAC: OME 20 mg, AMO 1 g, CLA 500 mg, bid	N/A	VA: 93.5% (86/92); OAC: 83.9% (73/87), *p* = 0.042	Diarrhea: OAC (10.3%); VA (3.3%)
										Nausea/vomiting: OAC (14.9%); VA (5.4%)
										Bloating: OAC (12.6%), VA (4.3%)
Chey et al. (2022, The United States and Europe) [26]	RCT	992	UBT or biopsy	UBT	14 days	VA: VPZ 20 mg bid, AMO 1 g tid	LAC: LAN 30 mg, AMO 1 g, and CLA 500 mg, bid; VAC: VPZ 20 mg, AMO 1 g, CLA 500 mg, bid	N-R VA: 78.5% (208/265), LAC: 78.8% (201/255), VAC: 84.7% (222/262); CLR-R VA: 69.6% (39/ 56), LAC 31.9% (23/ 72), VAC: 65.8% (48/73); All VA: 77.2% (250/324); LAC: 68.5% (226/330), VAC: 80.8% (273/338)	N-R VA: 81.2% (177/218), LAC: 82.1% (174/212), VAC: 90.4% (198/219); CLR-R VA: 79.5% (35 /44), LAC: 29.0% (18/62), VAC: 67.2% (39/58); All VA: 81.1% (215/265), LAC: 70.0% (194/277), VAC: 85.7% (240/280)	Total: VA: (29.9%); LAC (34.5%); VAC (34.1%)
										Diarrhea: VA (5.2%); LAC (9.6%); VAC (4.0%)
										Headache/dizziness: VA (1.4%); LAC (1.4%); VAC (2.6%)
										Dysgeusia: VA (0.6%); LAC (6.1%); VAC (4.3%)
										Nausea/vomiting: VA (2.3%); LAC (4.6%); VAC (2.0%)
										Abdominal pain: VA (1.4%); LAC (2.0%); VAC (1.2%)
Lin et al. (2022, China) [30]	RCT	230	UBT	UBT	7 days	H-VA: VPZ 20 mg bid, AMO 750 mg qid L-VA: VPZ 20 mg bid, AMO 500 mg qid	VAC: VPZ 20 mg, AMO 750 mg, CLA 500 mg, bid	H-VA: 63.5% (54/85); L-VA: 58.3% (49/84); VAC: 60.7% (37/61)	H-VA: 65.1% (54/83); L-VA: 66.2% (49/74); VAC: 64.9% (37/57)	Total: H-VA (16.90%); L-VA (13.20%); VAC (24.10%)
										Diarrhea: H-VA (1.2%); L-VA (0); VAC (1.6%)
										Abdominal pain: H-VA (2.4%), L-VA (1.2%); VAC (6.6%)
										Dizziness: H-VA (2.4%); L-VA (1.2%); VAC (0).
										Nausea/vomiting: H-VA (4.7%); L-VA (6.0%); VAC (3.3%)
										Skin rash: H-VA (1.2%); L-VA (0); VAC (0)

Abbreviations: RCT, randomized controlled trial; RUT, rapid urease test; C, Culture test; Ag, Anti-*H. pylori* antibody test; UBT, urea breath test; SAT, stool antigen test; VPZ, vonoprazan; AMO, amoxicillin; CLA, clarithromycin; PPI: proton pump inhibitor; OME: omeprazole; OAC, omeprazole triple therapy; LAC: lansoprazole triple therapy; LAN, lansoprazole; VA, vonoprazan-amoxicillin dual therapy; VAC, vonoprazan-amoxicillin-clarithromycin; triple therapy; H-VA, high-dose amoxicillin combined with VPZ; L-VA, low-dose amoxicillin combined with VPZ; bid, twice daily; tid, three times daily; qid, four times daily; ITT, intention-to-treat; PP, per protocol; N/A, not available, CLR-R: clarithromycin resistant, N-R: without antibiotic-resistant.

**Table 2 antibiotics-12-00346-t002:** Assessment of the risk of bias of included RCTs.

Authors	Year	Randomization Process	Deviations from Intended Interventions	Missing Outcome Data	Measurement of the Outcome	Selection of the Reported Result	Overall Bias
Chey et al. [26]	2022	Low risk	Low risk	Low risk	Low risk	Low risk	Low risk
Horii et al. [27]	2021	Low risk	Low risk	Low risk	Low risk	Low risk	Low risk
Suzuki et al. [28]	2020	Low risk	Low risk	Low risk	Low risk	Low risk	Low risk
Zuberi et al. [29]	2022	Low risk	Low risk	Low risk	Low risk	Some bias	Low risk
Lin et al. [30]	2022	Low risk	Low risk	Low risk	Low risk	Low risk	Low risk

**Table 3 antibiotics-12-00346-t003:** SUCRA-based efficacy ranking league matrix showing the comparative efficacies of the regimens included in this Network Meta-Analysis.

Overall		
VA	1.25 (0.77, 2.06)	0.49 (0.25, 0.87)
0.80 (0.49,1.29)	VAC	0.39 (0.18, 0.72)
2.04 (1.15, 4.07)	2.55 (1.39, 5.46)	PPI
RCTs		
VA	1.30 (0.77, 2.33)	0.49 (0.23, 0.96)
0.77 (0.49,1.29)	VAC	0.38 (0.16, 0.76)
2.04 (1.04, 4.42)	2.66 (1.32, 6.07)	PPI
14-day treatment		
VA	1.34 (0.45, 3.96)	0.49 (0.18, 1.16)
0.75 (0.25,2.24)	VAC	0.36 (0.13, 1.03)
2.07 (0.86, 5.66)	2.77 (0.97, 7.93)	PPI
Asia		
VA	1.20 (0.60, 2.64)	0.34 (0.08, 1.58)
0.83 (0.38, 1.68)	VAC	0.29 (0.05, 1.60)
2.86 (0.63, 13.29)	3.47 (0.62, 19.93)	PPI
High-dose amoxicillin		
VA	1.18 (0.56, 2.29)	0.47 (0.21, 0.97)
0.85 (0.44,1.78)	VAC	0.40 (0.17, 092)
2.13 (1.03, 4.86)	2.52 (1.09, 5.91)	PPI

VA, Vonoprazan dual therapy; VAC, vonoprazan triple therapy; PPI, PPI triple therapy; N/A, not available; RCTs, Randomized controlled trials.

**Table 4 antibiotics-12-00346-t004:** SUCRA-based efficacy ranking league matrix showing the comparative safety of the regimens included in this Network Meta-Analysis.

Treatment Discontinuation		
VA	0.83 (0.15, 3.17)	0.67 (0.05, 5.91)
1.21 (0.32, 6.78)	VAC	0.81 (0.08, 9.01)
1.49 (0.17, 19.09)	1.24 (0.11, 12.07)	PPI
Incidence of adverse events		
VA	1.45 (0.81, 2.77)	1.92 (0.92, 4.90)
0.69 (0.36, 1.24)	VAC	1.31 (0.56, 3.61)
0.52 (0.20, 1.08)	0.76 (0.28, 1.78)	PPI
Diarrhea		
VA	1.43 (0.75, 3.02)	2.79 (1.18, 7.34)
0.70 (0.33, 1.34)	VAC	1.97 (0.75, 5.22)
0.36 (0.14, 0.85)	0.51 (0.19, 1.34)	PPI
Abdominal pain		
VA	0.92 (0.28, 2.95)	1.52 (0.25, 9.64)
1.09 (0.34, 3.52)	VAC	1.68 (0.27, 11.21)
0.66 (0.10, 4.07)	0.60 (0.09, 3.65)	PPI
Bloating		
VA	0.68 (0.15, 2.94)	3.37 (0.59, 23.29)
1.48 (0.34, 6.67)	VAC	5.09 (0.52, 56.40)
0.30 (0.04, 1.70)	0.20 (0.02, 1.94)	PPI
Constipation		
VA	0.56 (0.13, 1.95)	N/A
1.76 (0.51, 7.46)	VAC	N/A
N/A	N/A	PPI

VA, Vonoprazan dual therapy; VAC, vonoprazan triple therapy; PPI, PPI triple therapy; N/A, not available.

**Table 5 antibiotics-12-00346-t005:** SUCRA values concerning the safety of therapeutic interventions against *Helicobacter pylori* included in this Network Meta-Analysis.

	VA	VAC	PPI
Treatment discontinuation	0.369	0.513	0.618
Incidence of adverse events	0.938	0.433	0.128
Diarrhea	0.931	0.531	0.374
Abdominal pain	0.57	0.642	0.288
Bloating	0.592	0.834	0.075
Constipation	0.178	0.822	N/A

VA, Vonoprazan dual therapy; VAC, vonoprazan triple therapy; PPI, PPI triple therapy; N/A, not available.

## Data Availability

Not applicable.

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
