# Peer review of "The Efficacy and Safety of Vonoprazan and Amoxicillin Dual Therapy for Helicobacter pylori Infection: A Systematic Review and Network Meta-Analysis"

_antibiotics, 2023, doi:10.3390/antibiotics12020346_

Round 1
Reviewer 1 Report
This systematic review and meta-analysis evaluates the efficacy and safety of vonoprazan-based dual eradication therapy for Helicobacter pylori infection. All sections of the article are well written and in general comply with the PRISMA-2020 recommendations. The results of the work are of practical interest to gastroenterologists.
I recommend adding a full bibliographic description of reference 11:
Malfertheiner P, Megraud F, Rokkas T, Gisbert JP, Liou JM, Schulz C, Gasbarrini A, Hunt RH, Leja M, O'Morain C, Rugge M, Suerbaum S, Tilg H, Sugano K, El-Omar EM; European Helicobacter and Microbiota Study group. Management of Helicobacter pylori infection: the Maastricht VI/Florence consensus report. gut. 2022 Aug 8:gutjnl-2022-327745. doi: 10.1136/gutjnl-2022-327745.
Author Response
Thank you for pointing this out. It was incomplete because this paper had not been published online when we submitted our paper. This reference has now been updated.
Reviewer 2 Report
Since H. pylori antibiotic resistance is a big and real problem around the world, all ways to combat this phenomenon are very valuable. In this context, the main aim of the article entitled “The efficacy and safety of vonoprazan and amoxicillin dual therapy for Helicobacter pylori infection: A systematic review and network meta-analysis” is up-to-date and scientifically important. As the Authors themselves note, the manuscript did not avoid certain limitations (including in particular the very limited final number of articles included). Nevertheless, for the reasons I have outlined above, and because of the very promising results of the vonoprazan therapy worldwide, the meta-analysis presented within the current manuscript is worth publishing.
Below I would like to present a list of amendments, the inclusion of which will help to improve the quality of the manuscript:
- - Please include italics when writing "Helicobacter pylori" OR “H. pylori” in all places in the manuscript (currently the spelling is completely random - once with it, once without)
- - “Only vonoprazan triple therapy and reverse hybrid therapy consistently achieved the desired 90% eradication rates” [lines 19 AND 53] -> Although it has been demonstrated in this article, this statement is too general and cannot be presented as a doctrinal statement. There are many other articles showing that both of these therapies do not achieve this threshold and, e.g., BQT does. (Please tone down this statement and add information about other therapeutic options that manage to exceed this indicator (indicating a total of 4-5 articles).
- - The abstract is missing a key statement that appears at the end of the current manuscript and is a valuable summary of this analysis – “For empiric therapies of H. pylori infection among treatment naïve patients, vonoprazan dual therapy, vonoprazan triple therapy, and PPI triple therapy did not achieve high cure rates (> 90%).”
- - “As a novel potassium-competitive acid blocker (P-CAB), vonoprazan (VPZ) produced …” [line 60] -> As a novel potassium-competitive acid blocker (P-CAB), vonoprazan (VPZ), produced …
- - Lines 292-294: It is worth adding one or two more sentences explaining the above phenomenon. It is worth pointing out that limiting the amount of taking broad-spectrum antibiotics decreases the disturbance of the human microflora.
- - Line 304 AND 308: the phrase “may influence” and “may be the preferred” should be changed to “influences” and “is preferred”
-
Author Response
Please include italics when writing "Helicobacter pylori" OR “H. pylori” in all places in the manuscript (currently the spelling is completely random - once with it, once without)
Sorry for this oversight and these have been corrected.
“Only vonoprazan triple therapy and reverse hybrid therapy consistently achieved the desired 90% eradication rates” [lines 19 AND 53] -> Although it has been demonstrated in this article, this statement is too general and cannot be presented as a doctrinal statement. There are many other articles showing that both of these therapies do not achieve this threshold and, e.g., BQT does. (Please tone down this statement and add information about other therapeutic options that manage to exceed this indicator (indicating a total of 4-5 articles).
Thank you for your comments. We have toned down this statement and added other therapeutic options for the management of H. pylori infection in first paragraph of the Introduction. We have also modified the Abstract accordingly to address these comments.
The abstract is missing a key statement that appears at the end of the current manuscript and is a valuable summary of this analysis – “For empiric therapies of H. pylori infection among treatment naïve patients, vonoprazan dual therapy, vonoprazan triple therapy, and PPI triple therapy did not achieve high cure rates (>90%).”
Thank you for your suggestions. We have added this statement to the Abstract.
"As a novel potassium-competitive acid blocker (P-CAB), vonoprazan (VPZ) produced …” [line 60] -> As a novel potassium-competitive acid blocker (P-CAB), vonoprazan (VPZ), produced …
Sorry for missing the comma. The correction has been made.
Lines 292-294: It is worth adding one or two more sentences explaining the above phenomenon. It is worth pointing out that limiting the amount of taking broad-spectrum antibiotics decreases the disturbance of the human microflora.
Thank you for your comments. We have added a sentence with a reference to explain the association between the use of broad-spectrum antibiotics and the potential impact on human microflora, causing diarrhea.
Line 304 AND 308: the phrase “may influence” and “may be the preferred” should be changed to “influences” and “is preferred”.
We have made the changes as suggested.
Reviewer 3 Report
This network meta analysis is based on a few studies three from Japan isone from Pakistan and one from US and Europe
The authors have done their best to compare like to like by including per protocol analysis rather than intention to treat which would be more valid
The duration of treatment varies and as the authors admit 14 days is better than 7 days
The dose of clarithromycin was low in the Japanese studies as older studies suggested better results with dual therapy with clarithromycin 500mgs twice a day
The authors emphasise the need for further studies
They should clearly state the studies required dose duration and comparators
How many of the studies quoted were Pharma sponsored?
Line 138 should read urea breath test not urine breath test
The value of the NMW Has to be questioned as there are so few studies all with different duration dose and populations
Antibiotic sensitivity should be addressed as classical triple therapy is obsolete if antibiotic sensitivity to clarithromycin is not performed
Since vonoprazin is not widely available Bismuth quadruple and non Bismuth quadruple which achieves greater than 90 percent eradication rates although mentioned needs to be highlighted
Author Response
This network meta-analysis is based on a few studies three from Japan is one from Pakistan and one from US and Europe.
Yes. That is correct.
The authors have done their best to compare like to like by including per protocol analysis rather than intention to treat which would be more valid.
Thank you for your comments.
The duration of treatment varies and as the authors admit 14 days is better than 7 days.
Yes. You are right. Thank you for your comments.
The dose of clarithromycin was low in the Japanese studies as older studies suggested better results with dual therapy with clarithromycin 500mgs twice a day.
We have added a sentence in Lines 147-150 to explain the doses of clarithromycin used in studies conducted in Japan and other countries.
The authors emphasize the need for further studies.
Yes.
They should clearly state the studies required dose duration and comparators
Thank you for your comments. The detailed information about doses, durations, and comparators have been included in Table 1.
How many of the studies quoted were Pharma sponsored?
Based on the full-text reading, only one study conducted in the US and Europe mentioned that it was sponsored by a pharmaceutical company, other studies did not provide this information. Thus, we are unable to definitely state how many studies were in fact sponsored by Pharma.
Line 138 should read urea breath test not urine breath test.
We are sorry for this minor mistake. It has been corrected.
The value of the NMW Has to be questioned as there are so few studies all with different duration dose and populations.
Thank you for your concerns. Compared to other meta-analyses on this topic, this NMW searched multiple databases and gathered all available RCTs that compared vonoprazan dual therapy versus vonoprazan triple treatment and/or PPI triple therapy. We performed subgroup analyses to detect the effect of different duration, doses, or populations on eradication rates. Thus, this NMW provides more complete analyses of the current literature to guide the management of H. pylori infection.
Antibiotic sensitivity should be addressed as classical triple therapy is obsolete if antibiotic sensitivity to clarithromycin is not performed.
We agree with your statement. Thus, we have stated that therapy should be guided by susceptibility testing in Lines 346 to 349.
Since vonoprazan is not widely available Bismuth quadruple and non-Bismuth quadruple which achieves greater than 90 percent eradication rates although mentioned needs to be highlighted
Thank you for your comments. Since the included studies did not compare available Bismuth quadruple therapy and non-Bismuth quadruple therapy with vonoprazan-based treatment, we are unable to make this statement in our manuscript.
Round 2
Reviewer 3 Report
The paper is fine Would like to see mentioned if triple therapy with clarithromycin is used in western population resistance should be known
Author Response
The paper is fine. Would like to see mentioned if triple therapy with clarithromycin is used in western population resistance should be known.
The use of vonoprazan triple therapy with clarithromycin should be avoided when resistance rate to clarithromycin is greater than 10%, and the local resistance rate should be known before starting vonoprazan triple therapy because the resistance rate to clarithromycin has been as high as 30% in some studies conducted in the US (Lines 362-366).